# Human Papillomavirus-Associated Oropharyngeal Cancer: Global Epidemiology and Public Policy Implications

**DOI:** 10.3390/cancers15164080

**Published:** 2023-08-13

**Authors:** Sifon Ndon, Amritpal Singh, Patrick K. Ha, Joyce Aswani, Jason Ying-Kuen Chan, Mary Jue Xu

**Affiliations:** 1Department of Otolaryngology-Head and Neck Surgery, University of California San Francisco, San Francisco, CA 94115, USA; 2School of Medicine, University of California San Francisco, San Francisco, CA 94143, USA; 3Department of Surgery, University of Nairobi, Nairobi 00100, Kenya; 4Department of Otorhinolaryngology, Head and Neck Surgery, The Chinese University of Hong Kong, Hong Kong SAR, China

**Keywords:** human papillomavirus, oropharyngeal cancer, gender-neutral vaccination policy

## Abstract

**Simple Summary:**

Head and neck cancers of the oropharyngeal subsite can be driven by the human papillomavirus (HPV). In countries such as the United States, the incidence of HPV-associated oropharyngeal cancer has exceeded that of HPV-associated cervical cancer. HPV vaccination is currently the main preventative approach for HPV-associated oropharyngeal cancer. Globally, both the epidemiology of HPV-associated oropharyngeal cancer and HPV vaccine policy vary. This paper aims to describe regional variations in HPV-associated oropharyngeal cancer, variations in gender-neutral vaccine policy, and future areas of policy-relevant research.

**Abstract:**

Global trends in human papillomavirus (HPV)-associated head and neck cancers (HNC), specifically in the oropharynx subsite, have been dynamically changing, leading to new staging and treatment paradigms. Epidemiologic studies have noted regional variations in HPV-associated oropharyngeal squamous cell carcinoma (OPSCC). While HPV vaccination remains the main preventative approach, vaccination policy in relation to gender neutrality is heterogeneous and particularly sparse in low- and middle-income countries, where the burden of global cancer cases and HPV-associated HNC are not well-characterized in certain regions. This review summarizes the existing literature on regional variations of HPV-associated OPSCC and gender-neutral vaccine policies. Based on available data, the incidence of HPV-associated OPSCC is highest in North America, Europe, and Oceania. As of 2022, 122 of 195 (63%) World Health Organization (WHO) member states had incorporated HPV vaccinations nationally; of these, 41 of 122 (34%) member states have introduced gender-neutral vaccine coverage. Future research is needed to describe continued evolving trends in HPV-associated OPSCC, understand underlying risk factors leading to regional variation in disease, and implement gender-neutral policy more broadly.

## 1. Introduction

The number of cases of human papillomavirus (HPV)-associated oropharyngeal cancers has surpassed that of cervical cancer in high-income countries such as the United States [1]. Across the globe, an estimated 30% of oropharyngeal squamous cell carcinomas (OPSCC), which primarily involve the tonsils and the base of the tongue, are driven by HPV, with HPV16 being the most common subtype associated with malignancy [2]. Multiple studies have shown regional and sex-based variations in incidence of HPV-associated OPSCC, with the highest incidences in North America, Europe, and Oceania [3,4]. Although the majority of studies suggest that HPV-associated OPSCC affects younger age groups [5], studies have also noted a rise in HPV-associated OPSCC among the elderly population over 70 years of age in countries such as the United States [3,6].

With the growing epidemic of HPV-associated OPSCC globally, there is an opportunity to expand the current coverage of gender-neutral HPV vaccinations. HPV vaccination is the best preventative approach but requires strategic implementation prior to sexual debut and exposure to sexually transmitted HPV to be most effective [7]. Studies have shown that HPV vaccinations have an efficacy of 88–93% against oral HPV infection [8,9]; however, gender-neutral HPV vaccination policy is not well-characterized relative to the global burden of HPV-associated OPSCC.

In this review, we examine the literature to characterize the epidemiology of HPV-associated OPSCC by region and compare the current policies for gender-neutral HPV vaccination. Characterizing gaps in both epidemiologic data and policies is needed to influence future areas of research and tailor public health interventions.

## 2. Global Epidemiology and Regional Variations of Human Papillomavirus-Associated Oropharyngeal Squamous Cell Carcinoma

### 2.1. Regional Variations of Human Papillomavirus-Associated Oropharyngeal Squamous Cell Carcinoma Incidence

The incidence of HPV-associated OPSCC varies both regionally and between sexes (Figure 1). Across studies, the age-standardized incidence rate (ASIR) has been highest in North America, Europe, and Oceania [2,4]. Additionally, while HPV-associated OPSCC affects both sexes, the ASIR is consistently higher in men compared to women [2,4]. Finally, globally, the attributable fraction (AF) or proportion of oropharyngeal cancers driven by HPV has ranged from 30.8% to 42.7% [2,4].

#### 2.1.1. North America

The North American region has the highest incidences of HPV-associated OPSCC in the world. The ASIR is estimated to be 3.41 per 100,000 in males and 0.71 in females, with an estimated 63% AF [4].

Particularly in the United States, the prevalence of HPV among OPSCC is an estimated 66.3% (CI 56.1–75.9); these trends have additionally been increasing [4,10,11]. The population-level incidence of HPV-positive oropharyngeal cancers increased by 225% (95% confidence interval 208–242%) from 1988 to 2004 [10]. During the early phase of the HPV epidemic for head and neck cancers, individuals diagnosed were typically younger, white men. However, over the past decade, the prevalence of HPV-associated OPSCC has also increased among older age groups, shifting the median age of presentation [3]. Additionally, from 1995–2012 at two academic hospitals, significant increases in the proportion of p16-positive OPSCCs occurred among women (from 29% to 77%; *p* = 0.005) in addition to men (from 36% to 72%; *p* < 0.001) [11]. Finally, from 1995–2012, the proportion of p16-positive OPSCCs increased significantly for race groups defined as nonwhite (from 32% to 62%; *p* = 0.02) in addition to white (from 39% to 86%; *p* < 0.001) [11].

Similar increasing patterns have been reported in Canada [12]. The proportion of tonsillar cancers that were HPV-positive substantially increased from 25% in 1993–1999 to 62% in 2006–2011 (*p* < 0.002) [13]. Similar observations of young nonsmokers with advanced nodal disease were also described [13]. A study conducted using the Canadian Cancer Registry reported that the ASIR of HPV-associated oropharyngeal cancer had increased significantly from 1.6 per 100,000 population in 1992 to 2.6 in 2009 [14]. The increase in OPSCC overall was more significant in males than in females, with an annual percentage change of 1.50% for men compared to 0.8% for women [15]. 

#### 2.1.2. Europe

The ASIR of HPV-associated OPSCC in Europe is an estimated 1.72 per 100,000 population among males and 0.41 among females, the third highest value regionally following North America and Oceania. Particularly in men, the ASIR is the highest in countries such as France (4.18) and Slovakia (3.36), while the overall ASIR among women is <1.00 across reported European countries [4]. 

In addition to incidence, the proportion of HPV-associated OPSCC also varies across Europe. An estimated 41.9% of all OPSCC cases in Europe are thought to be driven by HPV [4].

Based on data collected between 1990 and 2012, the proportion of HPV-associated OPSCC varies from approximately 50–70% among various European countries and is higher in Northern and Central-Eastern European countries than in Southern and Western European countries [3,16]. In a systematic review of data from seven European countries, the distribution of HPV positivity among patients with OPSCC varied from 18% to 65% between 2014 and 2018. Similarly, the highest proportion of patients with HPV positivity was observed in Northern European countries, specifically Sweden and Denmark, while the lowest proportion was observed in Greece and the Netherlands [17].

These trends have also been changing over a short period of time. Among OPSCC patients in Denmark, 62% of cases between 2011 and 2014 were HPV-positive [18]. This was increased compared to a previous study on HPV-associated tonsillar cancer in the same region from 2000 to 2010 [19].

#### 2.1.3. Asia

The ASIR of HPV-associated OPSCC is on the lower side regionally at 0.49 per 100,000 population in males and 0.10 in females. The AF in Asia is also overall lower compared to that of Western countries at 34.6% [4]. This trend persists in the most populated Asian countries such as India and China, where the proportion of OPSCC attributed to HPV infection ranges from 15–23% in India and 26–32% in China [4,20,21].

The region’s overall low incidence contrasts with countries in the region such as Singapore, in which the incidence of HPV-associated OPSCC is reported to be trending towards that of Western countries. [22] Assessing cases at a tertiary hospital, Fu et al. (2021) estimated the incidence for HPV-associated OPSCC in Singapore to range from 0.30 to 0.81 per 100,000 persons per year from 2015 to 2019 [22]. An additional study of archival tissue samples in Singapore reported the proportion of HPV-associated OPSCC to be 73.7% [23].

#### 2.1.4. Oceania

The incidence of HPV-associated OPSCC is second highest in Oceania, following North America. The ASIR for men in Oceania is 1.98 per 100,000, while the ASIR for women in Oceania is 0.42. Additionally, the regional AF (50.2%) is higher than the global average (42.7%) [4].

Studies from Australia report a general pattern of increasing prevalence of HPV-associated OPSCC. A study based upon the National Cancer Statistics Clearing House database (NCSCH) in Australia reported that from 1982 to 2005, there were annual increases in tonsil (1.39%, 95% CI 0.88–1.92%) and base-of-tongue cancers in males (3.02%, 95% CI 2.27–3.78%) [24]. Additionally, a PCR study targeting the E6 region and p16 in Australia reported an increase in the proportion of HPV association in OPSCC increased from 20% to 63% between 1987 and 2010 [25].

#### 2.1.5. Africa

The estimated regional ASIR is 0.44 per 100,000 population for males and 0.10 for females [4]. Of note, these ASIR estimates reported by Lu et al. extrapolated the AF of 58.5% from one study in South Africa for the region [4]. A recent systematic review in Sub-Saharan Africa reported that the p16 positivity among oropharyngeal cases was 20.3%, while the HPV PCR positivity was 15.1% among 31 studies representing 12 countries [26]. Future studies are needed to establish more accurate regional estimates.

#### 2.1.6. South/Central America

In Latin America, the ASIR per 100,000 population for males is 0.33 compared to 0.07 for females, the lowest compared to other regions [4]. Additionally, the proportion of OPSCC attributed to HPV ranges from 13–18% [2,4]. Of note, most of the available data on HPV-associated OPSCC in this region come from Brazil, as it is one of only a few countries in the region with a cancer registry. However, overall, data from the International Agency for Research on Cancer concluded that there did not appear to be a significant increase in oropharyngeal cancers in South America from 2000–2010 [27]. More studies would improve the epidemiologic understanding in the region.

### 2.2. Regional Variations of Oropharyngeal Squamous Cell Carcinoma Mortality

Globally, the age-standardized mortality rate for OPSCC overall is higher in men (0.89 per 100,000 population) compared to women (0.17) [12]. Age-standardized mortality rates based on available data are also higher in high-income countries (1.14 per 100,000 population for males and 0.22 for females) compared to low-income countries (0.80 in males and 0.51 in females). The mortality rates varied regionally but did not appear to follow a clear correlation to the incidence of OPSCC; mortality was highest in European males (1.70), while it was relatively lower in North American males (0.87), despite both regions being among those with the highest incidences of HPV-associated OPSCC [12].

### 2.3. Global Context for Human Papillomavirus Association Testing

Understanding of the history and current landscape of testing for HPV association in OPSCC contextualizes limitations in the existing data. Testing for HPV association has only recently been incorporated into guidelines for routine oncologic care and cancer staging within the past decade. The 2018 College of American Pathologists (CAP) guidelines for HPV testing in head and neck cancers proposed standardizing HPV testing across pathology practices [28]. The consensus recommended that (1) high-risk HPV testing be conducted for all new OPSCC patients (including lymphadenopathy with unknown primary) and (2) p16 immunohistochemistry can be used as a surrogate marker of HPV using 70% cytoplasmic/nuclear staining as a cutoff for positivity.

Later that year, the American Society of Clinical Oncology endorsed the CAP guidelines, clarifying a move towards use of specific terminology to distinguish between p16 versus high-risk HPV status when describing specimens [29]. Finally, the American Joint Commission on Cancer (AJCC) incorporated testing for HPV association and downstaging of HPV-associated oropharyngeal cancers in the eighth edition of its staging manual, given improved prognosis for this subset of patients [30]. These guidelines have led to the incorporation of HPV testing into clinical practice. A retrospective study published in 2020 of head and neck cancer referrals at Vanderbilt University in the United States demonstrated an increase in correct use of HPV testing for oropharyngeal cancer specimens from approximately 70% to 90% in the year before and after publication of the CAP guidelines [31].

Despite these recent changes in testing and staging for HPV-associated OPSCC, these changes have been studied mostly in the context of Western countries; however, the reliability of testing in different populations needs to be considered. Among the methods of testing for HPV association, p16 staining was the most inexpensive method with the highest sensitivity in a review by Augustin et al. (2020) [32]. However, the test utility may vary by population. A meta-analysis noted higher diagnostic efficacy of p16 testing in Western/European countries compared to other countries, with a combined diagnostic odds ratio of 69% for the non-Western countries [33]. The authors proposed that pathologist subjectivity may be a factor. Murthy et al. (2017) reported significant discordance between p16 and PCR positivity, particularly in India, with a high tobacco burden [34]. One mechanistic explanation is that this discrepancy may be attributed to p16 hypermethylation; lack of p16INK4a expression may be due to effects of various risk factors including heavy tobacco use leading to p16INK4a gene deletion or promoter methylation and loss of p16INK4a expression [34,35,36].

In addition to the quality and applicability of testing, the availability of testing limits the interpretation of epidemiologic data. Sub-Saharan Africa is a region in which testing and therefore epidemiologic data are limited. A survey of 16 fellowship-trained head and neck surgeons from 13 African countries reported that routine p16 testing is not available given the lack of testing facilities [37]. Given the resource limitations for testing of HPV association, it is unclear whether the latest AJCC/UICC staging system is appropriate for resource-constrained health systems which cannot routinely perform testing [38]. Variations in and lack of testing capacity influence the availability of epidemiologic data which can then further influence investment in HPV testing infrastructure and vaccination policies [26].

## 3. Human Papillomavirus Vaccination and Prevention of Human Papillomavirus-Associated Oropharyngeal Squamous Cell Carcinoma

Since the initial approval of the bivalent HPV vaccine in 2006 for the prevention of HPV-related diseases, both the number of HPV genotypes and indication of the HPV vaccine have expanded. Currently, there are six available vaccines that have been licensed globally: three bivalent vaccines, two quadrivalent, and one nonavalent. All vaccines provide protection against the high-risk HPV types 16 and 18, with additional coverage against types 6 and 11 provided by the quadrivalent vaccine and further coverage for the high-risk types 31, 33, 45, 52 and 58 conferred by the nonavalent vaccine [39]. Vaccine approval was predicated on demonstrating effective prevention of cervical precancerous lesions, genital warts, and anal neoplasia. Given that these vaccines all provide protection against HPV types 16 and 18, which are the subtypes implicated in approximately 85% of HPV-associated head and neck cancers [40], the United States Food and Drug Administration approved an expanded indication for the nonavalent HPV vaccine distributed by Merck (Gardasil 9) to include the prevention of HPV-associated oropharyngeal and other head and neck cancers in 2020 [41]. This vaccine is indicated in male and female children and adults ages 9 to 45.

Despite the presumed possibility of reduction in the incidence of HPV-associated head and neck cancers through vaccine prevention, there are no studies that provide direct evidence of this relationship. Existing studies examining the role of HPV vaccination in oropharyngeal cancers have used persistent oral infection as a surrogate marker of risk for cancer. This endpoint for vaccine effectiveness is likely based on the fact that viral DNA is diffusely present in tumor cells of HPV-associated cancers [42]. In one double-blind, randomized, controlled trial, it was found that women who received the bivalent HPV vaccine had a statistically significant reduction in oral HPV 16 and 18 infection four years following intervention [8]. A second randomized, controlled trial showed high vaccine efficacy for the prevention of persistent oral HPV infection, but low efficacy for single detection of any vaccine-type HPV at final visit. This study was terminated early due to futility [43].

Additional cross-sectional studies have suggested vaccine efficacy based on reduced prevalence of oral vaccine-type HPV in vaccinated individuals [9,44,45,46]. An additional longitudinal cohort study found that detection of vaccine-type HPV in oral rinse samples was significantly lower in vaccinated female adolescents compared to their unvaccinated counterparts [47]. Epidemiologic differences in oral HPV prevalence have also provided evidence in support of vaccine efficacy. In Europe, comparison of a young cohort and an older cohort who reached sexual maturity prior to and after licensing of the HPV vaccine in Europe, respectively, demonstrated a lower incidence of oral HPV DNA in the younger vaccinated group [48]. A phase-three randomized, double-blind, placebo-controlled clinical trial studying the efficacy of the nonavalent vaccine for prevention of persistent oral infection in adult males is currently underway, with the anticipated study completion date in 2024 [49,50].

## 4. Regional Variations in Vaccine Coverage and Policies

### 4.1. Human Papillomavirus Vaccine Coverage

As of 2022, 122 of 195 (63%) World Health Organization (WHO) member states have introduced HPV vaccinations nationally (Figure 2) [51]. Given the initial indications for cervical cancer, HPV vaccine policies have focused more on vaccination of eligible females. Eight years after the initial introduction of the HPV vaccine, 120 million women worldwide were reported to have been administered at least one dose of the HPV vaccine between 2006 and 2014 [52]. Globally, only 15% of females in the targeted age range are estimated to be fully vaccinated [53,54].

Only recently has there been increased implementation of gender-neutral vaccination strategies worldwide. Australia became the first country to introduce HPV vaccination for boys in 2007 [55]. In 2009 in the United States, the quadrivalent HPV vaccine was introduced for boys and integrated into the standard immunization schedule in 2011 for males aged 11–12 years [56,57]. As of 2022, 41 WHO nations had coverage for the first dose of an HPV vaccine for males by age 15 (Figure 3) [51,54]. An estimated 4% of boys had completed the entire series of the HPV vaccine, in comparison to 15% of girls, globally in 2019 [58]. While gender-neutral vaccination policies have expanded, there are still significant gaps at both the policy and implementation levels.

Additionally, there are disparities in HPV vaccine coverage between high-income countries (HICs) and low-income countries (LICs) [53,54,59]. After the HPV vaccine was initially introduced, it took less than 10 years for approximately 80% of HICs to adopt and implement the vaccine. In contrast, low- and middle-income countries (LMICs) not only started introducing the vaccine at a later stage but also at a slower pace [54]. According to 2020 estimates, approximately 85% of HICs had implemented HPV vaccination programs, whereas only 30% of LICs had achieved the same level of coverage [3]. In terms of the first dose of an HPV vaccine, while the global coverage in 2018 stood at 40%, LICs had only attained 10% coverage [53]. Continued, multidisciplinary efforts are needed to address financial, social, and health systems barriers to not only expand gender-neutral vaccine policies, but also policy and implementation plans adapted to lower income countries.

### 4.2. Barriers to HPV Vaccine Coverage

This observed trend of lower HPV vaccine coverage in low- and middle-income countries (LMICs) compared to high-income countries (HICs) can be attributed to several factors including cost and implementation strategies. In 2011, Gavi, the Vaccine Alliance and UNICEF, international public health partnerships and organizations, respectively, reached an agreement to provide HPV vaccines for USD 4.50 per dose for LICs [58]. However, the cost for middle-income countries (MICs) remained approximately three times higher than the negotiated price of HPV vaccines for Gavi-eligible countries [58,60]. Additionally, resource-constrained health systems sustaining recurrent HPV vaccination programs are associated with higher operational costs and resource needs, unlike mass single-dose or catch-up campaigns as in the case of measles and rubella vaccination [51,61,62,63]. The recent WHO recommendations for an alternative one-time dose schedule may mitigate this latter barrier [39,64].

Coverage has also been impacted by limitations in global supplies of vaccines. The introduction of Gavi support to LICs is contracted through only two manufacturers (GlaxosmithKline and Merck), which has outpaced their production capacity and impacted the supply of HPV vaccines to these regions [58]. Consequently, a global shortage of HPV vaccines, which is predicted to last until 2023 [60], will continue to potentially lead to delays in implementing catch-up campaigns and introducing the vaccines in LMICs in the coming years until the current vaccine manufacturers increase their production and additional manufacturers enter the market [54].

### 4.3. Innovations in Vaccine Delivery and Implementation

Innovative approaches in implementing the HPV vaccine globally have led to improved vaccine coverage. Vaccination administration programs in low-resource settings in schools are one approach to reaching adolescents, who tend to use health services to a lesser extent [65,66]. In Africa, all countries that reported coverage above 50% used school systems to deliver the vaccine [67]. In Rwanda, school-based strategies, implemented with a campaign-style approach, have resulted in a remarkably high HPV vaccine coverage rate of 94% among eligible girls as of 2019 [53,68]. School-based delivery strategies have also been shown to have increased success reaching males for vaccination [69,70].

In addition to delivery systems, the use of one- or two-dose vaccination schedules, has shown promise in reducing dropout rates [69,71]. Specifically in relation to the initially recommended three-dose schedule, a two-dose schedule was noted to be less expensive and to also allow for both doses to be delivered within the same school year, thereby maximizing coverage. Observational data from women who have received fewer than three doses have still shown vaccine efficacy against cervical cancer [72]. Barnabas et al. (2022) recently published their results on a randomized, multicenter, double-blind, controlled trial assessing the effect of a single dose of HPV vaccination compared with a meningococcal vaccine on the effect of persistent vaccine-type HPV infection at 18 months, measured on cervical and vaginal swabs in women [73]. The group noted a one-dose vaccine efficacy of 97.5%. An additional randomized trial in Tanzania assessing the antibody response following one, two, and three doses of either the bivalent or nonavalent vaccines had similar conclusions that single-dose HPV vaccination may provide adequate protection in the context of improved access and implementation [74]. While additional studies will be needed to understand the impact of a one-dose vaccine strategy on oropharyngeal HPV infection, these frameworks serve as innovative approaches to conceptualize and implement HPV vaccination in resource-constrained health systems.

### 4.4. Benefits of Gender-Neutral HPV Vaccination Policies

Studies have indicated that relying solely on girls-only HPV vaccination programs, which primarily aim to prevent cervical cancer and achieve herd immunity, may not effectively reduce the increasing incidence of HPV-associated OPSCC [67]. Female-directed programs require significantly higher and sustained coverage rates for a longer duration to eradicate persistent and carcinogenic HPV subtypes like HPV16 [67]. Model analysis has demonstrated that achieving 90% vaccination coverage among girls could potentially eliminate cervical cancer as a public health concern in most low- and middle-income countries (LMICs) within a century. However, in high-incidence countries with an age-standardized rate above 25 per 100,000, vaccination alone may not be sufficient to bring down the incidence of cervical cancer [75,76,77]. In contrast, gender-neutral vaccination programs with moderately high coverage rates (over 75%) could potentially go beyond reducing cervical cancer and even eliminate specific HPV vaccine types from circulation [76,77]. Moreover, female-only vaccination policies do not cover HPV transmission in men who have sex with men [18]; gender-neutral vaccination has shown a significant reduction of up to 69% in HPV transmission among this population group, as observed four years after implementing such programs [78].

**Figure 2 cancers-15-04080-f002:**
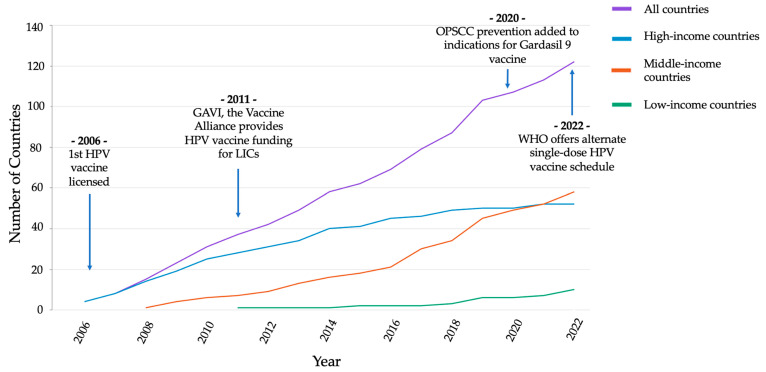
Number of Countries that Include HPV Vaccines as Part of Their National Vaccine Schedule per Year. GAVI, the Vaccine Alliance is a public-private global health partnership. HPV human papillomavirus. LIC low-income countries. OPSCC oropharyngeal squamous cell carcinoma. WHO World Health Organization. Data from the World Health Organization (WHO) immunization statistics [51].

**Figure 3 cancers-15-04080-f003:**
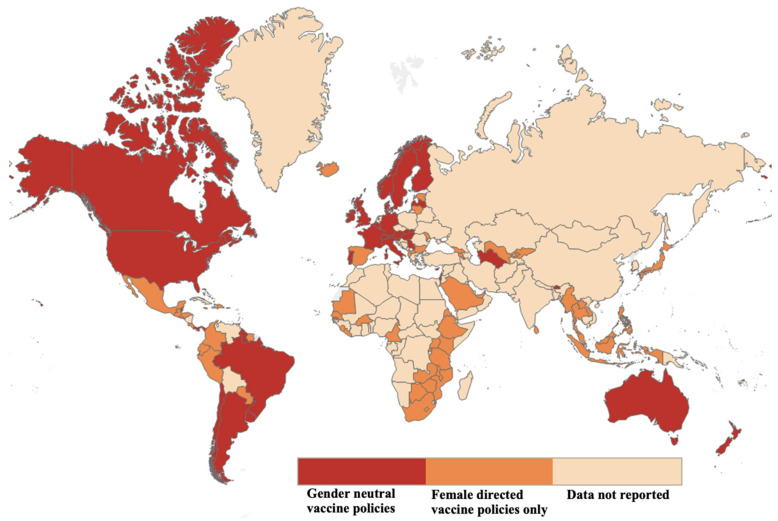
Female and gender-neutral vaccine policies globally. Data from the World Health Organization from 2022 [79].

## 5. Future Research Directions in Global Epidemiology and Policy-Relevant Research

The epidemiology of HPV-associated OPSCC is changing, and future research is critically needed to characterize the changing patterns and importantly to support the implementation of gender-neutral HPV vaccination. This review highlights the gaps in our understanding of the changing trends of HPV-associated OPSCC. Specifically, existing data emphasize the need for representative and robust sources of data; this will require increased availability of HPV histopathologic testing, investment in data collection systems such as cancer registries, and funding of studies from regions of the world in which data are currently sparse. Variations in available data also highlight an opportunity for increased HPV association testing at a clinical, patient-care level. Additionally, studies have even shown changes in trends over a 10-year span; as a result, continually updated data are also needed. Further research into understanding regional variations may additionally provide insight into the underlying biology and have implications for public health policies. Overall, the quality of data is central to policy recommendations and decisions.

In addition to a more comprehensive understanding of the changing epidemiology of HPV-associated OPSCC, there is a critical gap in public health policy research. While there has been implementation of HPV vaccination programs in over 122 countries/territories, only 41 countries/territories have gender-neutral vaccination policies [80]. Leveraging the changing trends of HPV-associated OPSCC for broader gender-neutral vaccination policies will benefit additional public health efforts around cervical and other genitourinary tract cancers.

## 6. Conclusions

There are regional trends in HPV-associated OPSCC that are changing and will require additional epidemiologic research moving forward. HPV vaccinations are the primary mode of prevention for HPV-associated OPSCC, and there exist gaps in the coverage and gender neutrality of HPV vaccination policies. Given the burden of HPV-associated malignancies beyond the head and neck, more efforts are needed to promote the global expansion and implementation of gender-neutral HPV vaccination programs.

## Figures and Tables

**Figure 1 cancers-15-04080-f001:**
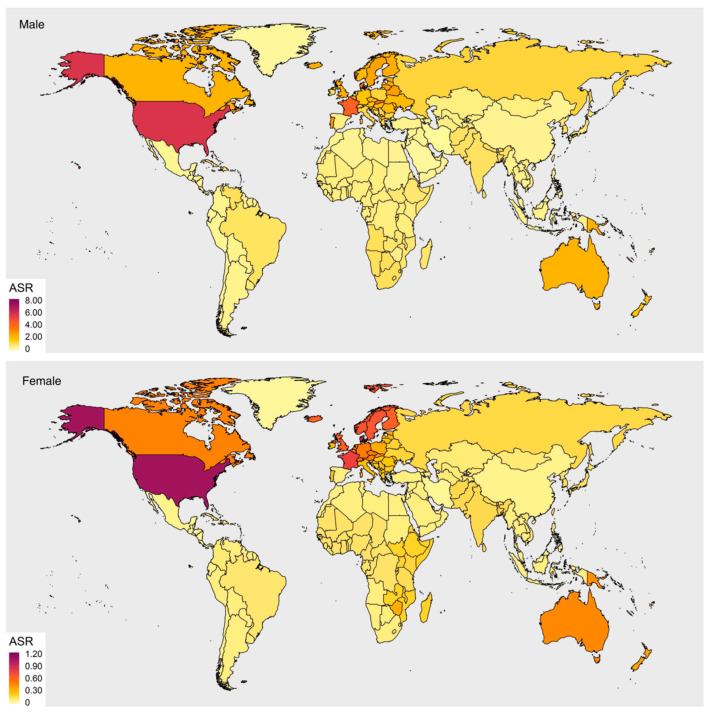
Age-standardized incidence rate (ASIR) of HPV-associated oropharyngeal cancer in males and females. Figure with permission from Lu et al. Lu Y, Xie Z, Luo G, Yan H, Qian HZ, Fu L, et al. Global burden of oropharyngeal cancer attributable to human papillomavirus by anatomical subsite and geographic region. *Cancer Epidemiology*. 2022; 78 [4]. ASR: age-standardized incidence rate.

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
