# Peer review of "Human Papillomavirus-Associated Oropharyngeal Cancer: Global Epidemiology and Public Policy Implications"

_cancers, 2023, doi:10.3390/cancers15164080_

Round 1

Reviewer 1 Report

The manuscript efficiently provides a comprehensive description of the epidemiology of HPV-related oropharyngeal cancers (OPSCC) as well as HPV vaccination policies in different geographic areas of the world. Given the recent increase in HPV-positive OPSCCs and the still not clear impact of HPV vaccination against head and neck cancers, it is imperative to carefully analyze these issues with a comprehensive approach. This goal is exhaustively addressed by the manuscript.

The document carefully addresses each topic discussed. However, the division into sections, especially the second paragraph "Global epidemiology and regional variations of human papillomavirus-associated oropharyngeal squamous cell carcinoma" is rather confusing and should be better organized. For example, the section about HPV testing guidelines should be placed after describing the epidemiology of HPV-positive OPSCC or should be treated as a separate paragraph.

Pag. 3, line 111: “2.2. Regional Variations of Human Papaillomavirus-Associated Ororopharyngeal Squamous Cell Carcinoma”

Pag. 7, line 294: “2.3. Regional Variations of Human Papaillomavirus-Associated Ororopharyngeal Squamous Cell Carcinoma”

These two paragraphs have the same title, the second should be corrected by referring to the topics covered in the corresponding section.

Pag. 3, lines 128-129: “The North American region overall has among the highest incidences of HPV-associated OPSCC in the world following Europe and Australia.”

This information is nowehere to be found in the document you cite as reference. It would be appropriate to add the correct citation.

Usage of the english language is great, just a few corrections are needed.

Reviewer 2 Report

The manuscript by Sifon Ndonet al, titled Human Papillomavirus-Associated Oropharyngeal Cancer: Global Epidemiology and Public Policy Implications, has shown Epidemiologic studies have noted regional variations in HPV-associated oropharyngeal squamous cell carcinoma (OPSCC). They have further shown regional variations of HPV-associated OPSCC and gender-neutral vaccine policies across the globe. The article is interesting in its current form but there are some suggestions to improve before publishing.

1.       The goal of research should be to implement clinical diagnosis and treatment of OPSCC. From the manuscript, we do not have a good understanding of how the author’s team implemented the clinical work. Can the results of this research approach lead to rapid clinical guidance?

2.       A graphical abstract showing variations in HPV-associated oropharyngeal squamous cell carcinoma (OPSCC) across gender, and the globe would give a clearer picture of the entire review.
